# Drivers of Dyadic Cofeeding Tolerance in *Pan*: A Composite Measure Approach

**DOI:** 10.3390/biology11050713

**Published:** 2022-05-06

**Authors:** Nicky Staes, Kim Vermeulen, Edwin J. C. van Leeuwen, Jonas Verspeek, Jonas R. R. Torfs, Marcel Eens, Jeroen M. G. Stevens

**Affiliations:** 1Behavioural Ecology and Ecophysiology Group, Department of Biology, University of Antwerp, 2610 Antwerp, Belgium; ejcvanleeuwen@gmail.com (E.J.C.v.L.); jonas.verspeek@kmda.org (J.V.); jonas.torfs@uantwerpen.be (J.R.R.T.); marcel.eens@uantwerpen.be (M.E.); jeroen.stevens@uantwerpen.be (J.M.G.S.); 2Centre for Research and Conservation, Royal Zoological Society of Antwerp, 2018 Antwerp, Belgium; 3Animal Behavior and Cognition, Department of Biology, Utrecht University, 3584 Utrecht, The Netherlands; 4SALTO Agro- and Biotechnology, Odisee University College, 9100 Sint-Niklaas, Belgium

**Keywords:** *Pan paniscus*, *Pan troglodytes*, tolerance experiment, relationship quality, principal component analysis

## Abstract

**Simple Summary:**

In this study on zoo-housed chimpanzees and bonobos, we propose a novel operational measure to quantify how tolerantly animals cofeed around a food source. Using an experimental set-up, we measured five behavioral variables that reflect positive and negative aspects of cofeeding behavior. Using a dimension reduction analysis technique, we found a two-dimensional tolerance model that we labeled “Tolerant Cofeeding” and “Agonistic Cofeeding”. In both species, scores on these two dimensions were predicted by sex, relatedness and social bonds in a similar fashion. High-value relationships between individuals that consist in mutual affiliation and support during aggressive interactions resulted in higher tolerance around food sources, while highly competitive dyads that show more mutual aggression showed lower tolerance. Bonobos and chimpanzees did not differ in their overall cofeeding tolerance, which likely reflects their adaptability to fluctuating social and ecological circumstances. This methodology can now be applied to other animal species to further investigate how variation in social and ecological environments influences the tolerance levels of social bonds, both at the dyadic and group level.

**Abstract:**

This study aimed to construct a composite model of Dyadic Cofeeding Tolerance (DCT) in zoo-housed bonobos and chimpanzees using a validated experimental cofeeding paradigm and to investigate whether components resulting from this model differ between the two species or vary with factors such as sex, age, kinship and social bond strength. Using dimension reduction analysis on five behavioral variables from the experimental paradigm (proximity, aggression, food transfers, negative food behavior, participation), we found a two-factor model: “Tolerant Cofeeding” and “Agonistic Cofeeding”. To investigate the role of social bond quality on DCT components alongside species effects, we constructed and validated a novel relationship quality model for bonobos and chimpanzees combined, resulting in two factors: Relationship Value and Incompatibility. Interestingly, bonobos and chimpanzees did not differ in DCT scores, and sex and kinship effects were identical in both species but biased by avoidance of the resource zone by male–male dyads in bonobos. Social bonds impacted DCT similarly in both species, as dyads with high Relationship Value showed more Tolerant Cofeeding, while dyads with higher Relationship Incompatibility showed more Agonistic Cofeeding. We showed that composite DCT models can be constructed that take into account both negative and positive cofeeding behavior. The resulting DCT scores were predicted by sex, kinship and social bonds in a similar fashion in both *Pan* species, likely reflecting their adaptability to changing socio-ecological environments. This novel operational measure to quantify cofeeding tolerance can now be applied to a wider range of species in captivity and the wild to see how variation in local socio-ecological circumstances influences fitness interdependence and cofeeding tolerance at the dyadic and group levels. This can ultimately lead to a better understanding of how local environments have shaped the evolution of tolerance in humans and other species.

## 1. Introduction

Tolerant food sharing among non-kin is considered a hallmark of human cooperative nature [1,2]. In its broadest sense, food sharing can be defined as the joint use of a food source and can take two forms: (1) offering food to another individual and (2) cofeeding, which is feeding in proximity around a food source and often requires tolerance to the proximity of others. In most animal species, food sharing occurs between kin [3,4,5] and mates [6,7,8] as the costs associated with sharing are directly offset by the fitness benefits gained through kin selection and any produced offspring [9]. However, humans belong to one of the few species that also regularly share food outside of the kin or mating context [1,2]. This raises an evolutionary question, since this type of food sharing does not provide obvious fitness benefits [10], but rather results in the loss of a resource or increased competition around the food source [1,2]. This contradiction can be explained by the concept of fitness interdependence, which is defined as the degree to which the fitness of two or more individuals is influenced by their interactions, both positive and negative [11,12]. Interactions among both kin and non-kin can thus result in higher fitness, for example, through cooperative hunting or increased protection against predators [13]. The level of tolerance around food sources would then reflect a balance between within-group competition on the one hand and fitness interdependence on the other, as individuals rely on each other for reproduction and survival [11,12,13,14].

In humans, the level of tolerant food sharing among non-kin is, among other things, dependent on the quality of the social relationship between the individuals participating in the transfer. For example, human children prefer sharing with a friend over sharing with a familiar child that is not a friend [15]. Non-human primates can develop very similar close long-lasting social relationships with unrelated individuals that are referred to as friendships [16,17,18], and similar to humans, high-quality social bonds predict more tolerant sharing [19,20,21,22,23]. However, social bonds in these studies are typically portrayed by single measures such as grooming or proximity that often fail to capture the complexity of primate social relationships [21,24,25,26]. A multidimensional model of dyadic relationship quality is available for primates and is based on dimension reduction methods such as factor analysis that consider a variety of behavioral characteristics of the relationship but reduce them to fewer dimensions in an objective manner [27,28,29,30]. The benefit of this method is that both positive and negative aspects of social relationships are taken into consideration while keeping the number of components that will be tested low, which increases power of statistical analyses in small sample sizes inherent to many primate behavioral studies.

Here, we investigated the role of relationship quality on Dyadic Cofeeding Tolerance in zoo-housed bonobos (*Pan paniscus*) and chimpanzees (*Pan troglodytes*). Given that they are the two primate species most closely related to humans [31], both are keystone species for investigating the mechanisms underlying the evolutionary origins of human cofeeding tolerance [32,33,34]. Similar to humans, food sharing in the *Pan* species is not limited to kin or mates, but also takes place between unrelated group members [6,10,35,36,37], which is rare among primates and makes bonobos and chimpanzees good models to investigate the impact of both kin and non-kin fitness interdependency interactions on tolerance [38]. Interestingly, previous studies on cofeeding tolerance have reported contrasting results in *Pan*. While in general, bonobos are portrayed as more peaceful and egalitarian than chimpanzees [35,39,40,41,42,43,44], this does not always result in higher tolerance during cofeeding tests [36,45,46,47,48,49]. Results from cofeeding studies are difficult to generalize though, as differences in results could be due to the variety of methodologies used (e.g., dyadic cofeeding or codrinking tests, measures of food transfers in a group setting, cofeeding tests in a group setting [35,46,48,50,51]), or the fact that within-species variation in cofeeding tolerance might be greater than between-species variation [45,52,53]. Multigroup studies using identical experimental set-ups in both species are thus warranted but are currently lacking. Finally, for both species, multidimensional relationship quality models were successfully constructed in the past that show great overlap between the two species [16,28,29,30]. Two factors of relationship quality were consistently found in both species: Value and Incompatibility. Relationship Value refers to the benefits an individual gains from his social partner, such as grooming or agonistic support [16,28,29,30]. The second component, Incompatibility, is a measure of the general nature of the interactions between the two partners and typically reflects levels of aggression and counter-intervention during agonistic interactions. In both species, measures of relationship quality were influenced by variables such as sex, kinship, age difference, familiarity and personality [16,28,29,30] (for an overview see Table 1).

Previous work in *Pan* demonstrated that chimpanzees shared more food with group members with whom they had more valuable relationships, and that bonobos tolerated more food transfers from group members with whom they had more affiliative relationships [36]. However, these relationship quality components were constructed for a small sample of 11 chimpanzees and 6 bonobos, and the number of behavioral variables entered in the factor analysis was limited compared to the later, more elaborate chimpanzee and bonobo relationship quality models [28,29,30]. Therefore, in this study, we included a larger multigroup sample of zoo-housed chimpanzees (N = 22) and bonobos (N = 21). In contrast to previous studies, we constructed a de novo relationship quality model by analyzing identical behavioral variables for both species and including individuals from both species in the same factor analysis, hereafter referred to as “the *Pan* model” as both species are combined into one model. This ensured that behavioral loadings on measures of relationship quality were comparable for bonobos and chimpanzees, and that scores of both species could be included in the same statistical model to test for potential species differences on the relationship between cofeeding tolerance and relationship quality. To validate the model, we compared the resulting *Pan* relationship quality model to previously described models for chimpanzees and bonobos and tested how scores on components differed between dyads of different sex combinations, age differences and kin relationships. If the resulting components of our *Pan* model were valid, we predicted that sex, age and kinship influenced relationship quality in species-specific patterns, as shown in previous studies (Table 1).

Second, we propose a novel approach to quantify a multidimensional model for Dyadic Cofeeding Tolerance by using factor analysis to construct composite measures that consider both positive and negative food interactions [14]. In this way, we aimed to capture diverse aspects of cofeeding tolerance that could associate differently with specific relationship quality components. While measures of relationship quality were assessed based on dyadic behavioral data observed in a naturalistic condition throughout the apes’ daily lives [28,29,30], cofeeding tolerance was quantified using a validated group cofeeding paradigm: the resource plot experiment [45,52]. In this paradigm, a group of animals was provided with a desirable, non-monopolizable food resource that was adjusted to the group size in a designated feeding area in their enclosure. In previous studies, cofeeding tolerance was typically determined at the group-level by quantifying a single measure: the proportion of the group cofeeding in the plot [45,53,54]. Here, we used the same paradigm to determine novel composite tolerance measures at the dyadic level to identify what dyadic strategies might underly previously reported group differences in tolerance [45,53,54]. The benefit of this approach is that it takes into account multiple facets of complex social relationships while reducing the number of variables to avoid overfitting of models on datasets with relatively small sample sizes. To do so, we scored five dyadic measures of positive and negative food-related behaviors in and around the feeding area during the experiment: proximity in the feeding area, frequency of aggression, frequency of tolerant food transfers, frequency of negative food-related behavior and presence of a dyad in the feeding area [14]. To allow for direct comparison of the results between bonobos and chimpanzees, an identical methodology was used in all groups of both species.

The aim was to test if scores on dimensions of Dyadic Cofeeding Tolerance differed between species, between dyads with different sex combination, age difference, maternal kinship and relationship quality scores. We expected that bonobos would not systematically differ from chimpanzees, in line with previous results using similar feeding experiments, and that sex, age and kinship effects on cofeeding tolerance would follow species-specific patterns similar to what is shown for relationship quality components. We expected to find an association between measures of Dyadic Cofeeding Tolerance and relationship quality similar to previous studies [8,19,21,26,36], with higher cofeeding tolerance present in dyads with higher Relationship Values and lower Incompatibilities. Since Relationship Value was consistently higher in maternal kin dyads in both *Pan* species [28,29,30], the resulting higher fitness interdependence in these dyads was predicted to result in higher Dyadic Cofeeding Tolerance. For dyads with high Incompatibility and thus high levels of competition [28,29,30], interdependence should be lower, and Dyadic Cofeeding Tolerance was expected to be lower.

## 2. Methods

### 2.1. Subjects and Housing

We studied two groups of chimpanzees housed at Safaripark Beekse Bergen in Hilvarenbeek (The Netherlands) and three groups of bonobos, of which two were housed in Frankfurt Zoo in Frankfurt-am-Main (Germany) and one in Zoo Planckendael in Mechelen (Belgium). Experimental and behavioral data on the ape groups were collected by three observers between August and December 2019. In total, the subjects included 21 bonobos (7 males, 14 females), whose ages ranged from 7 to 68 years, and 22 chimpanzees (10 males, 12 females), ranging from 9 to 38 years old. All groups contained infants and/or juveniles (<7 years), but following the methodology of previous studies on relationship quality these were excluded from analysis given that the nature of their social relationships substantially differs from those of mature individuals, especially when considering levels of grooming and agonistic behavior [28,29,30]. Enclosure sizes differed between the zoos, but the apes had access to an indoor and outdoor enclosure in all facilities, containing installations for climbing as well as various enrichment items. Food was offered at least three times a day and consisted mainly of a mixture of vegetables, fruits and pellets. Water was available ad libitum in all zoos. Details on group compositions and observations can be found in Table 2.

### 2.2. Data Collection

#### 2.2.1. Relationship Quality

In each group, we collected behavioral data in a naturalistic setting (i.e., throughout the apes’ daily lives) to determine dyadic relationship quality. Focal animal sampling, group scans of proximity and all occurrence sampling of agonistic behaviors were conducted following an institutional ethogram for social behavior that was identical for both species [47,55]. Before starting the data collection, all observers were subjected to rigorous training for at least two weeks and tested for inter-observer reliability by scoring a focal video. Inter-observer reliability reached a mean of *r* = 0.84 across all observers, meaning that all observations were highly reliable [56]. All behaviors were scored live using The Observer (Noldus version XT 14, The Netherlands). In total, the data collected included 408.45 h of focal observations (mean 9.50 h per individual), 2217 group scans (mean 443 scans per group) and 127.62 h of all occurrence observations (mean 25.50 h per group) (Table 2).

Measures of relationship quality were then determined for both species in an identical manner, following the methodology described in Stevens et al. [30]. Although this model of relationship quality was originally used in bonobos, we combined both chimpanzees and bonobos in the same model to increase comparability of the results between the species. The bonobo model closely resembles the chimpanzee relationship quality models by Fraser et al. [29] and Koski et al. [28] on which it is based, but it was constructed using a more conservative statistical procedure and fewer behavioral variables, leading to a two-factor model. From the observational data, we extracted dyadic scores for eight social behavioral variables: grooming frequency, grooming symmetry, proximity, aggression frequency, aggression symmetry, support, counter-intervention and peering (for definitions of the behavioral variables, see Appendix A). The eight variables were corrected for observation time, square-rooted to improve normality of the data and entered into a dimension reduction analysis.

#### 2.2.2. Dyadic Cofeeding Tolerance

To measure cofeeding tolerance, we tested each group of apes in an experimental paradigm called the “resource plot”, which is based on the methods of a group cofeeding paradigm previously performed in a zoo-housed group of bonobos [45]. In this assay, each group is provided with a desirable non-monopolizable food source that has the possibility to elicit competition and aggression, and the apes’ interactions around the food are recorded. However, instead of using peanuts as in the previous study, we used cooked pasta because some bonobos in our current sample were known to have peanut allergies. To take differences in group size into account, the quantity of pasta and the surface area of the plot were scaled to the size of the group. The number of pasta pieces delivered to each group was determined by multiplying the group size by 12 (excluding individuals < 3 years). The area over which the pasta was distributed (hereafter referred to as the “resource zone”) was 1 m wide for all groups, but the length of the plot was calculated in such a way that the density of pasta equaled 60 pieces per m^2^.

Experimental sessions took place between 09.00 and 11.00 h. Before the start of a session, the apes were confined to their indoor quarters, and a keeper familiar to the animals showed them a transparent bag containing the pasta. Once the apes had seen the food, the experimenter distributed the food evenly over the resource zone in the apes’ outdoor enclosure (Figure 1a). Ten minutes after showing them the pasta, the apes were released (Figure 1b) and their behavior was recorded using two video cameras (Legria HFR88, Canon, Tokyo, Japan). One camera was focused on the resource zone and a ± 2 m edge around it, and the other camera was used to film the behavior of individuals outside the plot, when in sight. Recording continued until all apes retreated back inside or until all pasta was consumed, which usually took 2 to 3 min. Each group was tested in three habituation sessions and eight experimental sessions, with the exception of bonobo group 1 at Frankfurt, which was tested in five instead of eight experimental sessions due to bad weather conditions. The habituation sessions were used to familiarize the apes with the set-up prior to the actual test but were not included in the analyses. Only one session was conducted per day, and all sessions were completed within one month. Sessions did not coincide with other feedings or keeper interaction times, and no sessions were conducted on rainy days.

To measure cofeeding tolerance, we coded from the videos of the experimental sessions each individual’s social interactions occurring within one arm’s length (±1 m) of the resource zone. All videos were coded in The Observer (Noldus version XT 14, The Netherlands). Coding started when the first ape arrived within one arm’s length of the resource zone and continued until all pasta was consumed or taken from the resource zone to be eaten elsewhere. Each individual in the group was observed using focal follows for the duration of the experiment. During these focal follows, we scored social and cofeeding behaviors occurring within the resource zone and combined them into three behavioral categories: frequency of aggression, frequency of tolerant food transfers and frequency of negative food-related behavior. The first category, frequency of aggression, contained all instances of aggressive interactions (aggressive intentions, directed displays, charges and pestering) within a dyad. The second category, frequency of tolerant food transfers contained all instances of positive food-related (collect near and relaxed claim) interactions within a dyad. Finally, frequency of negative food-related behavior combined all instances of intolerant behavior around the food (food shielding, stealing and displacements) within a dyad. For an overview of the behaviors included in each behavioral category and their respective definitions, see Table 3. To make the frequency data comparable between groups, we divided the number of occurrences of each behavioral category by the total duration of the experimental sessions for that group.

Simultaneously with focal observations, group scans were carried out to determine for each individual’s proximity to others in the resource zone. Scans started at 15 s intervals for 2 min after the first ape arrived at the resource zone. Proximity was scored following three categories: “close” (≤1 m of another individual); “touch” (physically touching another individual); and “alone” (>1 m of another individual). Proximity data were then used to calculate two additional dyadic variables: being together in the resource zone and being close in the resource zone (Table 3). The variable being together in the resource zone represents the proportion of scans that a dyad was seen together in the resource zone regardless of whether they were in close proximity or not at the time of the scan, divided by the total number of scans taken for that group. Finally, the variable being close in the resource zone represents the proportion of scans a dyad spent within arm’s reach in the resource zone (thus either touching or within 1 m of another individual), divided by the number of times that dyad was seen together in the resource zone during a scan.

These five variables (frequency of aggression, frequency of tolerant food transfers, frequency of negative food-related behavior, being together in the resource zone and being close in the resource zone) were scored similarly in both species, and then entered in a factor analysis to create a composite measure model for Dyadic Cofeeding Tolerance.

### 2.3. Data Analysis

#### 2.3.1. Dimension Reduction Analysis

We then performed a factor analysis with varimax rotation and Kaiser normalization to obtain composite measures of relationship quality and Dyadic Cofeeding Tolerance. Visual analysis of the scree plot and parallel analysis were used to determine the number of factors to extract [57]. Parallel analysis compares eigenvalues from the dataset to eigenvalues from random matrices of a similar sample size [58]. Only factors exceeding the 95th percentile of values derived from random matrices are retained [58,59]. Coefficients of correlations of the behavioral variables ≥ |0.4| were considered salient [57]. Factor and parallel analyses were conducted using IBM SPSS 25. As we based our behavioral variable selection for relationship quality modeling on the bonobo model [16,17,18,19,20,21,22,23,24,25,26,27,28,29,30], we used the same methodology to construct a model using just the chimpanzee data to ensure that there were no major species differences in factor item loadings. This model was highly similar to the *Pan* relationship quality model, with the exception of peering no longer loading on either of the two dimensions (see Appendix A).

#### 2.3.2. Factors Affecting Relationship Quality and Dyadic Cofeeding Tolerance

We ran linear and general linear mixed models with the lmer and glmer functions (lme4 package 3.6.3 [60]) in R (version 4.1.2) to assess if relationship quality and Dyadic Cofeeding Tolerance components differed between species, or between dyads with different kinships, sex combinations and/or age differences. In these models, we treated each component as a response variable in a separate model. The variables species (chimpanzee or bonobo), kinship (maternal kin or not maternal kin), sex combination (male–male, female–male, female–female) and an interaction term between species and sex combination were considered as categorical predictor variables in each of these models. Age difference was added as a continuous variable to each model. As individuals occurred across multiple dyads, we added identity of the two individuals forming the dyads as random effects to account for interdependence of the data. We also corrected for group effects by adding group as a third random effect, which was in turn nested within “species” as each group was either bonobo or chimpanzee but never both. Dyads were classified as related when the average coefficient of relatedness exceeded 0.125 through the maternal line [30]. Kinship was then entered into the model as a binary variable. This resulted in seven mother-offspring dyads and one half-sibling dyad in bonobos. In chimpanzees, we identified one aunt–niece dyad, one uncle–nephew dyad and one grandmother–offspring dyad. For sex combination, our sample included 58 female–female dyads, 92 female–male dyads and 26 male–male dyads. These numbers were lower for our tolerance models due to exclusion of dyads not seen entering the plot together (see below). Age differences were calculated by taking the absolute difference of the years of birth for the two members of the dyad. For our tolerance models, we also included the two relationship quality components as predictor variables to test for effects of relationship quality on tolerance. Due to the low number of aggressive interactions during the period of observation, dominance ranks could not be reliably estimated and were therefore excluded from our models. Normality and homogeneity of variances of the outcome variables were first confirmed with diagnostic plots (QQ-plots and residuals vs. fitted). We used likelihood ratio tests to test for fixed effects using the *drop1* function in R with the argument test set to “Chisq”. Using this backwards selection approach, all non-significant effects were dropped until a final model was retained with only significant variables. Categorical variables with significant effects were further analyzed post hoc using the emmeans function with Tukey adjustment, from the package “emmeans” [61].

Given that a considerable number of dyads never entered the resource zone together, the tolerance data contained a high number of zeros and were not normally distributed. Therefore, we did a two-step analysis. First, we investigated if the likelihood that a particular dyad would be seen together in the resource zone was dependent on their relationship quality. To do so, each dyad was assigned a binary variable based on whether the individuals of the dyad were both seen in the resource zone at the same time (observed in the resource zone together at least once = 1; never observed in the resource zone together = 0). Then, we ran a generalized linear mixed model (GLMM) with a binomial error distribution and logit link function, with the binary variable as the response variable, and both factors of relationship quality as predictor variables in the same model. We also tested for general effects of species, age difference, sex combination, kinship and an interaction between species and sex combination by including these variables as predictor variables alongside the relationship quality factors. The random effect structure again consisted of the identity of the individuals forming the dyad and group nested within species. The significance of the full model was tested against a null model that only contained the random effects using a likelihood ratio test [62]. Only when this comparison yielded a significant result was the full model further explored.

Second, we investigated potential associations between our factors of relationship quality and factors of Dyadic Cofeeding Tolerance for those dyads that did spend time together in the resource zone using the linear mixed model approach outlined for relationship quality. In these models, only dyads that were observed together in the resource zone were included, resulting in a sample size of 109 dyads. Because we only included dyads that were in the resource zone together for these models, our sample consisted of 36 female–female, 52 female–male, 17 male–male and 4 mother–son dyads. Dyadic Cofeeding Tolerance factors were log-transformed to adhere to assumptions of normality. We used the grubbs test to remove outliers from the dataset from the package “outliers” in R. In the grubbs test, *p*-values < 0.05 indicate that the lowest or highest data point is likely to be an outlier and should thus be removed.

## 3. Results

### 3.1. Relationship Quality

We extracted eight behavioral variables from daily behavioral observations and entered these in a factor analysis to determine a joint *Pan* relationship quality model. Kaiser’s measure of sampling adequacy (KMO) was 0.651, which is acceptable, and Bartlett’s test of sphericity revealed that inter-variable correlations were sufficiently high (χ^2^ = 711.10, df = 28, *p* < 0.001). Using the scree plot and a parallel analysis, two factors were extracted (Table 4). The first factor explained 31.87% of the total variance and had high positive loadings for proximity, grooming frequency, peering and grooming symmetry. Therefore, we labeled this component “Relationship Value”. The second factor explained 20.81% and contained high positive loadings for aggression frequency, counter-intervention and aggression symmetry. We named this factor “Relationship Incompatibility”. An additional model run on just chimpanzees revealed an identical structure to the *Pan* relationship quality model with the exception that peering no longer loaded on any of the factors (Appendix A).

### 3.2. Factors Affecting Pan Relationship Quality

No significant species by sex combination interaction effect was found on either component, and component scores also did not differ between species or with age difference of the dyad (Table 5, Appendix A). Instead, Relationship Value and Incompatibility were significantly affected by kinship and sex combination of the dyad in a similar fashion in both species (Figure 1). Dyads that were maternal kin scored higher on Relationship Value (*p* = 0.014) and lower on Relationship Incompatibility (*p* < 0.001) compared to other dyads (Figure 2). Female–female dyads scored higher on Relationship Value (*p* = 0.015) and lower on Relationship Incompatibility (*p* < 0.001) than female–male dyads (Figure 2). Compared to male–male dyads, female–female dyads also scored lower on Relationship Incompatibility (*p* < 0.001) but not on Relationship Value (*p* = 0.636). Finally, female–male dyads did not differ from male–male dyads in either dimension (Table 5).

### 3.3. Dyadic Cofeeding Tolerance Model

Next, we tested all bonobo and chimpanzee groups in a cofeeding paradigm to construct composite measures for Dyadic Cofeeding Tolerance. Aggressive interactions during the feeding tests occurred a total of 26 times for both species combined (chimpanzees: n = 13; bonobos: n = 13). Tolerant food transfers happened frequently but were more common in chimpanzees (collect near: chimpanzees n = 66; bonobos n = 1); and relaxed claim: chimpanzees n = 1; bonobos n = 0). Intolerant food-related behaviors were rare but occurred more in chimpanzees (food-shielding: chimpanzees n = 6; bonobos n = 0, stealing: chimpanzees n = 0; bonobos n = 2, displacements: chimpanzees n = 1; bonobos n = 1). All five variables were then entered into a factor analysis to construct composite measures of *Pan* Dyadic Cofeeding Tolerance. Sampling adequacy was acceptable (KMO = 0.621), and inter-variable correlations were sufficiently high (Bartlett’s test of sphericity: χ^2^ = 151.76, df = 10, *p* < 0.001). Visual inspection of the scree plot and a parallel analysis yielded two factors (Table 6). The first factor explained 42.36% of the total variance and contained high positive loadings for being together and being close in the resource zone, as well as tolerant food transfers. Dyads that scored high on this factor thus represented dyads that entered the resource zone together, stayed more often in proximity and showed more tolerant food-related behavior. Therefore, we labeled this first factor Tolerant Cofeeding. The second factor explained 24.04% of the total variance and had high positive loadings for aggression and negative food-related behavior. Dyads that scored high on this factor thus engaged in conflict more often and showed more negative, intolerant food-related behavior, so we labeled this factor Agonistic Cofeeding.

### 3.4. Variables Affecting Dyadic Cofeeding Tolerance

Out of 176 dyads, only 109 (62%) were observed at least once in the resource zone during the paradigm, and this number was similar in chimpanzees (n = 66; 60%) and bonobos (n = 43; 65%, see Table 7). Due to the relatively large number of zeros present in the dataset, we performed a two-step analysis to investigate what variables affect Dyadic Cofeeding Tolerance in *Pan*. In step 1, we compared dyads that were observed in the resource zone against dyads that never entered the resource zone to see if they differed in their relationship quality, sex combination, age difference or kinship. In step 2, we investigated, for those dyads that did enter the resource zone together, what factors influence variation in their two composite measures of Dyadic Cofeeding Tolerance: Tolerant Cofeeding and Agonistic Cofeeding.

#### 3.4.1. Variables Affecting Likelihood of Being Observed in the Resource Zone (Yes/No)

The likelihood that dyads were observed in the resource zone was independent of species (χ^2^ = 1.956, *p* = 0.376), sex combination (χ^2^ = 0.347, *p* = 0.951), kinship (χ^2^ = 0.035, *p* = 0.982), age difference (χ^2^ = 2.306, *p* = 0.316) or species-by-sex interaction effects (χ^2^ = 2.762, *p* = 0.500). Dyads that were observed in the resource zone were those dyads with higher Relationship Value (χ^2^ = 13.959, Est = 1.387, SE = 0.42, *p* < 0.001; Figure 3) and higher Relationship Incompatibility (χ^2^ = 11.723, Est = 0.913, SE = 0.30, *p* = 0.003; Figure 3) compared to dyads that were never seen in the resource zone during the sessions.

#### 3.4.2. Variables Affecting Composite Measures of *Pan* Feeding Tolerance

For the Tolerant Cofeeding factor, one female–male dyad was first deleted from the dataset as it was clearly an outlier (G = 3.171, *p* = 0.011), resulting in 108 remaining dyads for testing. Tolerant Cofeeding did not differ between species, between dyads with larger or smaller age differences or with kinship and did not have significant species-by-sex interaction effects, indicating that sex effects on Tolerant Cofeeding were not species-specific (Table 5). Instead, across species, it significantly differed between dyads of different sex combinations (Figure 4) and Relationship Value (Figure 4). Male–male dyads scored significantly higher on Tolerant Cofeeding than female–female dyads (*p* = 0.021) and female–male dyads (*p* = 0.014). Female–female dyads did not differ significantly from female–male dyads (*p* = 0.838). Dyads with higher Relationship Value scored higher on Tolerant Cofeeding (*p* = 0.016), while no effect was found for Relationship Incompatibility (Table 5).

The second component of Dyadic Cofeeding Tolerance, Agonistic Cofeeding, did not differ between species, with age difference, with different sex combinations, Relationship Value, and was not influenced by species-by-sex interaction effects (Table 5). Instead, it significantly differed with levels of Relationship Incompatibility (*p* < 0.001) and kinship (*p* = 0.005) (Figure 4). Dyads with higher Relationship Incompatibility scored higher on Agonistic Cofeeding, and related dyads also scored higher on Agonistic Cofeeding than unrelated dyads (in both species (Table 5).

## 4. Discussion

This study aimed to construct a composite model of Dyadic Cofeeding Tolerance in bonobos and chimpanzees using a validated experimental cofeeding paradigm and to investigate whether components resulting from this model differ between the two species, or vary with factors such as sex, age, kinship and social bond strength. To do so, we also constructed a novel two-factor relationship quality model (Relationship Value and Incompatibility) for bonobos and chimpanzees combined, which enabled us to test for the effects of species and relationship quality dimensions on cofeeding tolerance in one model. Our Dyadic Cofeeding Tolerance model analysis resulted in two composite measures: Tolerant Cofeeding and Agonistic Cofeeding. Interestingly, there were no significant differences between the two species in Tolerant Cofeeding or Agonistic Cofeeding scores, and the effects of sex, kinship, Relationship Value and Relationship Incompatibility on Dyadic Cofeeding Tolerance were identical in both species.

### 4.1. Pan Relationship Quality Model

Our first aim was to construct a multidimensional relationship quality model for the two *Pan* species combined to obtain dyadic scores on relationship quality components that can be compared between species in further testing for relationship quality effects on Dyadic Cofeeding Tolerance. We found a two-factor model (Relationship Value and Incompatibility) identical to the previously found bonobo model [16,30], with the exception that support no longer loaded on either of the factors as the frequencies observed were also very low in our study. Similarly, no items related to security of relationships were used in this study as they were not available for all groups [30]. Compared to previous chimpanzee models [28,29], the two factors were also highly similar, with the exception that for the first component, Value, peering behavior replaced begging behavior, and for the second component, Incompatibility, tolerance to approaches was not scored and therefore not included in our study. As we based our behavioral variable selection on the published bonobo model, it is not surprising that our solution resembles the bonobo model more than the chimpanzee model. To investigate if the model would look similar if bonobos were excluded, we ran a separate model including only chimpanzee behavioral items and found a highly similar solution to the *Pan* model, with the exception that peering no longer loaded on Relationship Value (Appendix A). Why this is the case requires further investigation, as peering behavior was not less frequently observed in chimpanzees than bonobos. The lack of a correlation of peering with grooming and proximity in chimpanzees likely suggests that this type of behavior is differently distributed in chimpanzees than bonobos and likely serves a different purpose. The function of peering remains unclear in both species but has received more attention in bonobos than chimpanzees [55,63,64]. While some suggest that it might be a request for social tolerance [55], it is often performed in contexts where the recipient of the peering holds an interesting resource (food, material, infant) or behavioral commodity (e.g., grooming, object manipulation). Here, in bonobos, peering occurred more frequently in dyads with a high Relationship Value [30] whereas in chimpanzees, this appears not to be the case.

With this exception in mind, the results of our *Pan* relationship quality model indicate that the behavioral variable selection is suitable for relationship quality determination in both species and allows for between-species comparisons of relationship quality scores in further modeling of relationship quality effects on Dyadic Cofeeding Tolerance or on any other factors of interest in future studies. Both factors also showed relatively good validity, as shown by links with kinship and sex combination. In both species, kinship and sex effects on relationship quality followed identical patterns (Appendix A), with maternal kin having higher Value and lower Incompatibility scores and female–female dyads having higher Relationship Value and lower Incompatibility scores than female–male dyads, in line with previous findings in both bonobos and chimpanzees [28,29]. In contrast to what we expected, the species by interaction effect was not significant, and this likely explains why male–male and female–female dyads did not differ significantly from each other in Relationship Value, as contrasting sex effects on social bond strength were present in bonobos versus chimpanzees. Visual inspection of the plots (Appendix A) does indicate that in chimpanzees, male–male bonds had higher Value than any other dyad, while in bonobos female–female dyads and male–male dyads scored similarly high on Value, and both scored higher than female–male dyads. However, the number of male–male dyads was low in bonobos, and these results should thus be interpreted with caution. For Relationship Incompatibility, male–male dyads had higher Incompatibility than female–female dyads in both species. However, in contrast to previous chimpanzee findings, female–male dyads did not differ significantly from male–male dyads in Incompatibility, which is more in line with findings from bonobos [28,29].

### 4.2. Dyadic Cofeeding Tolerance

Our next aim was to construct a Dyadic Cofeeding Tolerance model based on novel composite measures using the resource plot paradigm [14]. Unlike previous studies that rely on single measures to represent tolerance, we combined five behavioral variables into a dimension reduction analysis, resulting in a two-factor Dyadic Cofeeding Tolerance model: Tolerant Cofeeding and Agonistic Cofeeding. Partners of dyads scoring high on Tolerant Cofeeding are more likely to jointly enter the resource plot, stay in close proximity while feeding in the plot and have more tolerant food sharing, while dyads scoring high on Agonistic Cofeeding behaved more aggressively towards each other during the experiments and showed more intolerant food-related behavior such as stealing and shielding.

Interestingly, we found no significant species differences in Tolerant or Agonistic Cofeeding. This finding adds to existing claims in the literature that bonobos are not necessarily more tolerant than chimpanzees in feeding contexts [36,45,46,47,48,49,54] and that between-group variation is likely larger than between-species variation [53,54,65]. Additionally, surprisingly, sex effects on both Tolerant Cofeeding or Agonistic Cofeeding did not differ for chimpanzees and bonobos. For both species, male–male dyads scored higher on Cofeeding Tolerance than any other dyad, but these results must be interpreted with caution as the effects are largely driven by chimpanzee data. Since for this part of the analysis dyads that never entered the resource zone were excluded, the sample size and thus power of the analysis significantly decreased, and only one male–male bonobo dyad was ever seen together in the resource zone, while in chimpanzees, 24% of dyads seen in the resource zone were male–male, indicating a clear sex bias in participation during the experiment. While male–male bonds can be highly social, tolerant, cooperative (but also competitive) in chimpanzees, in bonobos, relationships among males are typically weak and agonistic [66,67]. Male bonobo dyads also typically do not share food or give each other agonistic support [36,67]. In contrast, female bonobo dyads were expected to score higher on Tolerant Cofeeding and lower on Agonistic Cofeeding than any other dyad and therefore score more similar to chimpanzee male–male bonds, but species-specific sex effects on composite measures of Dyadic Cofeeding Tolerance were not significant in our study. However, when inspecting the percentage of dyads that participated and thus entered the resource zone for each sex combination, the patterns did match expectations (Table 7). In chimpanzees, 80% of male–male dyads entered the resource zone, while in bonobos, only 17% did. In contrast, only 40% of chimpanzee female–female dyads entered the resource zone while 86% of bonobo female dyads did. For female–male dyads, an intermediate score of about 60% was found for entering the plot for each species. We suspect that the lack of significant species-by-sex interaction effects on entering the plot is thus due to the relatively low sample size in each category. Future studies using larger sample sizes could help to resolve this issue.

Further results show that both tolerance measures were significantly predicted by components of relationship quality and that the effects were identical in both species. It is important to note is that behavior during the experiments was recorded and analyzed independently from behavior in the naturalistic setting. Any correlations between measures of Dyadic Cofeeding Tolerance and relationship quality were thus not due to repeated use of the same data but rather a reflection of similar social strategies in different contexts. Following our prediction, we found that dyads with higher Relationship Value, were more likely to enter the resource zone and score higher on Tolerant Cofeeding compared to dyads with lower Relationship Value. That is, individuals were more likely to enter the resource zone with partners with whom they also had stronger affiliative relationships outside of the experiments and to stay in close proximity to them and allow for more tolerant food transfers. These results highlight the importance of high Value relationships to reduce the cost of feeding competition. Competition for food access during a cofeeding test can lead to social tension and aggression, which in turn increases the risk of injury [68,69,70,71,72]. Maintaining valuable social bonds with high levels of mutual grooming can help mediate these tense social situations and allow for higher tolerance and closer proximity in these dyads during feeding [21,23,25,73,74]. This finding is consistent with work in other primate species, where more tolerant cofeeding and food sharing were found between partners with strong grooming bonds or partners that spend a lot of time in close proximity outside of feeding contexts [19,20,21,22,23].

In contrast to our expectations, more incompatible dyads were also more likely to enter the resource zone together. In other words, individuals also entered the resource zone with partners with whom they have a more agonistic, competitive relationship outside of the experiments (indicated by higher levels of mutual aggression and counter-intervention). However, even though they entered the resource zone together, they scored higher on Agonistic Cofeeding within the plot, in line with our predictions. These results can likely be explained by the nature of the food source. Since the resource plot could not be monopolized by a single individual, apes with varying levels of dyadic relationship quality are able to enter the resource zone [75,76]. Thus, while the even distribution of the food in the experiments allowed the apes to seek out closely affiliated partners to feed alongside, this same distribution also allowed individuals with more competitive relationships to enter the resource zone together. Altogether, these results demonstrate that social relationships outside of competitive feeding contexts can predict tolerance around valuable food resources. More specifically, dyads with high Relationship Value and low Incompatibility will cofeed more tolerantly and less agonistically.

Overall, our study highlights the potential of composite measure constructs to investigate complex social traits such as cofeeding tolerance and the factors driving variation in these traits across species. By combining specific behavioral traits from different species into one model, the resulting dimensions reduce the number of tests needed and allow for direct between-species comparisons as the behavioral loadings on each dimension are identical for both species but scores on the dimensions vary. This method does come with the limitation that not all behaviors serve identical functions in all species and as a result will drop out of the analysis, as was the case for peering behavior in our study.

## 5. Conclusions

Our results show that bonobos and chimpanzees do not differ consistently in Dyadic Cofeeding Tolerance, which supports earlier findings of within-species variation in cofeeding tolerance being larger than between-species variation in bonobos and chimpanzees [45]. It is important to note that the lack of clear species differences in Dyadic Cofeeding Tolerance might be attributable to the reduced importance of fitness interdependence in captive *Pan* populations due to more similar socio-ecological conditions versus wild populations. In the wild, bonobos and chimpanzees are both male philopatric species and experience similar predation risks [77,78]. However, resource distribution and food availability are believed to be responsible for differences in between- and within-group competition and levels of kin-driven fitness interdependence [77,78]. Generally speaking, in chimpanzees, seasonality is higher, and resources are more clumped, which results in higher competition between females, who each occupy and compete for high-quality core areas and show lower tolerance towards immigrating non-kin females and commit female infanticide [79,80]. Chimpanzee males are related because of male philopatry and form largely kin-driven alliances and strong social bonds to defend a communal territory with its resident females against males from neighboring communities. This results in strong between-group competition, including border patrols, intercommunity killing, raids and infanticide [81,82,83]. In bonobos, seasonality in food availability is generally considered to be lower and resources are less clumped, resulting in lower between-and within-group competition [41,84]. Relationships between bonobo communities are more relaxed; no intercommunity killing or infanticide has been described, and resident bonobo females are less hostile towards immigrating females [85]. Bonobo females form alliances with non-related females and form coalitions that allow for female codominance [41,86]. While male bonobos are related to each other because of male philopatry, similar to chimpanzee males, bonds between male bonobos are weak and they do not form coalitions or border patrols [66,67]. This shows that compared to chimpanzees, fitness interdependence in wild bonobos is less kin-driven. At a group level, this is expected to result in higher fitness interdependence in wild bonobo populations, and therefore higher levels of overall group cofeeding tolerance compared to wild chimpanzee populations.

Despite this reported species duality, bonobos and chimpanzees seem to show behavioral plasticity. Western chimpanzee populations seem to experience less food seasonality, resulting in more relaxed relationships between females compared to eastern populations [87], and within wild bonobo communities, between-group competition increases when food abundance is lower [88], and some populations seem to experience more seasonality in food availability, but how this relates to tolerance within and between groups is unknown [89]. This flexibility within *Pan* species likely explains why our study, as well as other captive work, did not find the predicted species differences in cofeeding tolerance, as resource distribution in captive populations is similar and intergroup competition is lacking, which reduces the need for male alliance formation in zoo-housed chimpanzees.

The quality of social bonds also predicted Dyadic Cofeeding Tolerance in an identical manner in both *Pan* species and provides a mechanism that could explain previously reported between-group variation in feeding tolerance in *Pan* [45,52], as the strength of social relationships can vary between populations depending on demographics [52] and ecological conditions [79]. Our results indicate that cofeeding tolerance in *Pan* is flexible and likely regulated through changing levels of mutual interdependence relationships, which are in turn reliant on environmental conditions. This flexibility in tolerance appears to outweigh species-specific physiological restraints on tolerance and highlights the adaptiveness of the two species to changing environments. Our study on zoo-housed chimpanzees and bonobos offers an operational measure to quantify cofeeding tolerance that can be applied to a wider range of populations of chimpanzee and bonobo populations, as well as other primate or mammal species, both in the wild and in zoos or sanctuaries, to see how variation in local socio-ecological circumstances influences fitness interdependence and cofeeding tolerance at the dyadic and group levels. This may ultimately lead to a better understanding of how local environments have shaped the evolution of tolerance in humans and other species.

## Figures and Tables

**Figure 1 biology-11-00713-f001:**
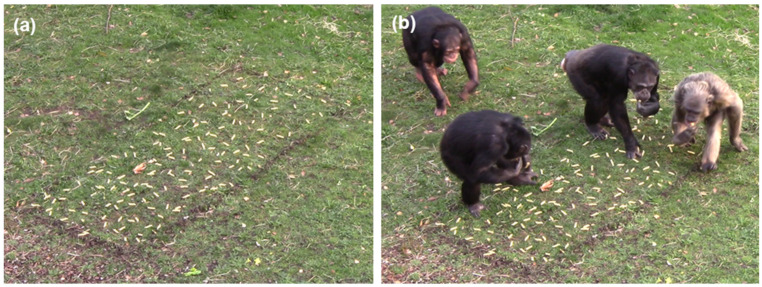
The pasta plot paradigm. (**a**) Still image from video showing the pasta distributed over the resource zone in the apes’ outdoor enclosure. (**b**) Still image from video shortly after the apes were released into their outdoor enclosure. These images show the chimpanzees of group 1 at Beekse Bergen Safaripark.

**Figure 2 biology-11-00713-f002:**
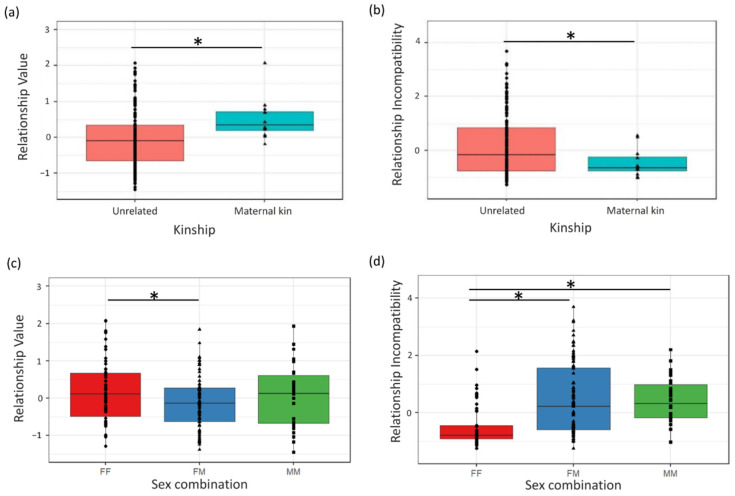
Factors affecting composite measures of *Pan* Relationship Quality. Dyads of maternal kin scored higher on (**a**) Relationship Value and lower on (**b**) Relationship Incompatibility compared to unrelated dyads. Female–female (FF) dyads scored (**c**) higher on Relationship Value than female–male (FM) dyads and (**d**) lower than FM and male–male (MM) dyads on Relationship Incompatibility. Boxplot figure with lower and upper box boundaries at 25th and 75th percentiles, respectively. Line inside box shows median, black dots show raw data for bonobos while black triangles indicate raw chimpanzee data points. * indicates significant at *p* < 0.05 level.

**Figure 3 biology-11-00713-f003:**
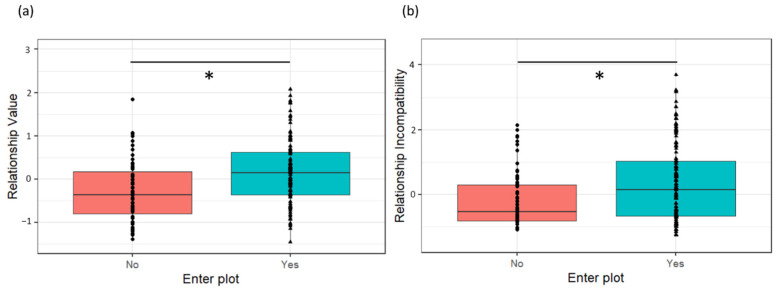
The likelihood that a dyad is observed in the resource zone is dependent of its (**a**) Relationship Value and (**b**) Relationship Incompatibility. “Yes” indicates that the dyad was observed at least once in the resource zone, whereas “no” indicates it was never seen. Boxplot figure shows lower and upper box boundaries at 25th and 75th percentiles, respectively. Line inside box shows median, black dots show data points falling outside 10th and 90th percentiles. * indicates significant at *p* < 0.05 level.

**Figure 4 biology-11-00713-f004:**
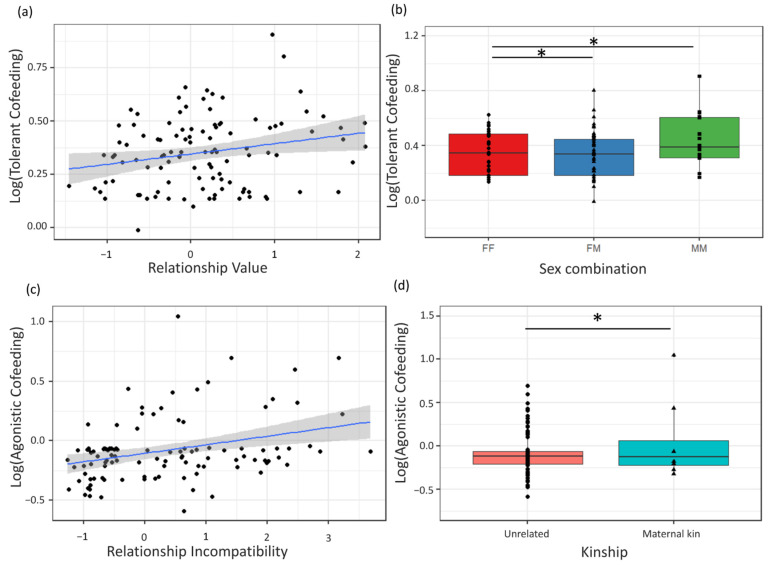
Factors affecting composite measures of *Pan* Dyadic Cofeeding Tolerance. Scores on Tolerant Cofeeding were higher in dyads with (**a**) higher Relationship Value and in (**b**) male–male (MM) dyads versus all other sex combinations (FF = female–female; FM = female–male). Scores on Agonistic Cofeeding were higher in dyads with (**c**) higher Relationship Incompatibility and (**d**) dyads with maternal kin compared to unrelated dyads. Boxplot figures (**b**,**d**) show lower and upper box boundaries at 25th and 75th percentiles, respectively. Line inside box shows median, black dots show all data points for bonobos while black triangles indicate all chimpanzee data points. * indicates significant at *p* < 0.05 level.

**Table 1 biology-11-00713-t001:** Overview of effects of kinship, sex combination and age difference on Relationship Value and Incompatibility in bonobos and chimpanzees from existing literature.

		Bonobo	Chimpanzee
Relationship Value	Kin vs. non-kin	>[30]	>[28,29]
	Sex (MF vs. FF)	<[30]	>[28]
	Sex (MM vs. FF)	<[30]	>[28]
	Sex (MF vs. MM)	<[30]	
	Large vs. small age difference	>[30]	<[29]
Relationship Incompatibility	Kin vs. non-kin		<[29]
	Sex (MF vs. FF)	>[30]	>[28,29]
	Sex (MM vs. FF)	>[30]	>[28,29]
	Sex (MF vs. MM)		>[29]
	Large vs. small age difference		

MF = male–female, FF = female–female, MM = male–male. Blank spaces indicate no significant effect was found. > indicates larger than, < indicates smaller than.

**Table 2 biology-11-00713-t002:** Details on group composition, time of behavioral data collection and observers.

Species	Group	Males	Females	Immatures	Mean Focal	Group Scans	All Occ	Year Observed	Observers
Chimpanzees	BB1	5	6	1	7.6	517	4.1	October–December 2019	KV
	BB2	5	6	3	7.6	525	4.2	August–October 2019	KV
Bonobos	FR1	3	3	3	13.0	425	7.3	August–October 2019	JT
	FR2	1	5	2	12.9	380	8.4	August–October 2019	JT
	PL	3	6	4	8.9	370	61.1	August–October 2019	IF & JT

Social groups: BB, Safaripark Beekse Bergen; FR, Frankfurt Zoo; PL, Zoo Planckendael. Numbers indicate distinct groups at the same site (only BB and FR). Immatures were individuals younger than 7 years. Mean focal indicates mean number of hours of focal observations carried out per group. Group scans indicate the total number of group scans performed per group. All occ indicates total number of hours of all occurrence group observations performed per group.

**Table 3 biology-11-00713-t003:** Behavioral variables, with corresponding definitions, scored during the experiments to determine measures of Dyadic Cofeeding Tolerance.

Behavioral Variable	Definition
Frequency of aggression	The frequency of all aggressive interactions (aggressive intentions, directed displays, charges and pestering) within a dyad. That is, the sum of all interactions from A to B and B to A.
Frequency of tolerant food transfers	Frequency of all instances of collect near (=subject waits for discarded food pieces, which are collected within arm’s reach of the receiver) and relaxed claim (=subject takes away food from receiver in a relaxed manner, without protest from the receiver) within a dyad. That is, the sum of all interactions from A to B and B to A.
Frequency of negative food-related behavior	Frequency of all instances of food shield (=subject positions himself between receiver and a food item apparently desired by the receiver), steal (=subject grabs food from receiver and then runs away while carrying the food) and displacement (=subject approaches receiver and forces receiver to leave) within a dyad. That is, the sum of all interactions from A to B and B to A.
Being together in the resource zone	Proportion of scans a dyad was seen together in the resource zone regardless of whether they were in close proximity or not at the time of the scan, divided by the total number of scans taken for that group.
Being close in the resource zone	Proportion of scans a dyad spent within arm’s reach in the resource zone (thus either touching or within 1 m of the another individual), divided by the number of times that dyad was seen together in the resource zone during a scan.

**Table 4 biology-11-00713-t004:** Varimax rotated factor loadings for the factors of the *Pan* relationship quality model.

Variable	Relationship Value	Relationship Incompatibility
Proximity	**0.877**	0.159
Grooming frequency	**0.869**	0.143
Peering	**0.646**	−0.069
Grooming symmetry	**0.507**	0.196
Support	0.365	−0.122
Aggression frequency	−0.041	**0.844**
Counter-intervention	−0.030	**0.742**
Aggression symmetry	0.198	**0.689**
% of variation explained	31.87%	20.81%
Eigenvalue	2.55	1.67

Boldface indicates loadings ≥ |0.4|.

**Table 5 biology-11-00713-t005:** Overview of variable effects on relationship Value, Relationship Incompatibility, Tolerant Cofeeding and Agonistic Cofeeding in bonobos and chimpanzees.

		Est	SE	F	*p*
Relationship Value	Kinship	0.540	0.210	6.116	**0.014**
	Sex (MF vs. FF)	−0.333	0.135	3.928	**0.015**
	Sex (MM vs. FF)	0.098	0.218	3.928	0.636
	Sex (MF vs. MM)	0.236	0.162	3.928	0.150
	Species	−0.387	0.233	0.788	0.383
	Age difference	−0.013	0.006	3.366	0.068
	Species * sex	/	/	1.050	0.354
Relationship Incompatibility	Kinship	−0.952	0.278	11.694	**<0.001**
	Sex (MF vs. FF)	0.959	0.168	4.230	**<0.001**
	Sex (MM vs. FF)	0.959	0.240	1.965	**<0.002**
	Sex (MF vs. MM)	−0.001	0.169	16.316	0.999
	Species	0.146	0.440	0.001	0.980
	Age difference	0.005	0.007	0.359	0.550
	Species * sex	/	/	0.616	0.542
Tolerant Cofeeding	Relationship Value	0.05	0.029	5.961	**0.016**
	Relationship Incompatibility	−0.011	0.015	0.617	0.435
	Kinship	−0.103	0.064	0.085	0.052
	Sex (MF vs. FF)	0.007	0.024	3.457	0.838
	Sex (MM vs. FF)	0.114	0.048	3.457	**0.021**
	Sex (MF vs. MM)	−0.107	0.035	3.457	**0.014**
	Species	−0.075	0.056	0.009	0.529
	Age difference	0.002	0.001	0.077	0.064
	Species * sex	/	/	0.040	0.394
Agonistic Cofeeding	Relationship Value	−0.0296	0.03145	0.54	0.464
	Relationship Incompatibility	0.087	0.021	17.722	**<0.001**
	Kinship	0.2027	0.096346	8.343	**0.005**
	Sex	/	/	1.355	0.264
	Species	0.160637	0.1053	0.89	0.387
	Age difference	0.0026	0.002306	1.529	0.219
	Species * sex	/	/	1.412	0.248

Est = estimate, SE = standard error, F = test statistic, *p* = *p*-value. Boldface indicates significant associations. For categorical predictors that were not significant, no further estimates are shown, as indicated by/. * indicates interaction effect between two variables.

**Table 6 biology-11-00713-t006:** Varimax rotated factor loadings for the factors of Dyadic Cofeeding Tolerance in chimpanzees and bonobos.

Variable	Tolerant Cofeeding	Agonistic Cofeeding
Being in close proximity in the resource zone	**0.883**	0.084
Being together in the resource zone	**0.74**	0.004
Frequency of tolerant food transfers	**0.707**	0.24
Frequency of aggression	−0.005	**0.878**
Frequency of negative food-related behavior	0.24	**0.828**
% of variation explained	42.36%	24.04%
Eigenvalue	2.12	1.2

Boldface indicates item loadings >|0.4|.

**Table 7 biology-11-00713-t007:** Overview of the total number of potential dyads per species across all groups observed and the number of dyads that entered the resource zone during the cofeeding paradigm. Dyads are shown by sex combination and in between brackets is the percentage of dyads seen in the resource zone out of the total number of dyads available for each sex combination.

	Potential Dyads in Total Sample				Dyads Seen Together in the Resource Zone			
Species	FF	MF	MM	Total	FF	MF	MM	Total
Chimpanzees	30	60	20	110	12 (40%)	38 (63%)	16 (80%)	66 (60%)
Bonobos	28	32	6	66	24 (86%)	18 (56%)	1 (17%)	43 (65%)

FF = female–female, MF = male-female, MM = male–male.

## Data Availability

The data presented in this study are available in Appendix A.

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
