# Peer review of "Drivers of Dyadic Cofeeding Tolerance in Pan: A Composite Measure Approach"

_biology, 2022, doi:10.3390/biology11050713_

Round 1
Reviewer 1 Report
The manuscript reports an important test to investigate DCT. Showing the similarity for the two species is interesting as it allows us to conclude that the evolutionary origin was shared by a common ancestor between humans and Pan!
Apart from the scientific value the study will also be a chance for animal keepers to improve their feeding protocols!
Author Response
Point 1: The manuscript reports an important test to investigate DCT. Showing the similarity for the two species is interesting as it allows us to conclude that the evolutionary origin was shared by a common ancestor between humans and Pan! Apart from the scientific value the study will also be a chance for animal keepers to improve their feeding protocols!
Response 1: We thank the reviewer for this positive feedback on our submission and we agree that the implications of this study are also relevant for husbandry purposes.
Reviewer 2 Report
The authors investigate Dyadic Cofeeding Tolerance in bonobos and chimpanzees using the “resource plot” experimental paradigm. The authors present a methodological innovation by constructing a novel Relationship Quality index. The study shows that DCT affected in the same way in both species. Finally, the authors conclude that this effect seems to be evolutionarily conserved and likely to be present in the common ancestor of humans and chimpanzees.
My main problem is the lack of theoretical background when deriving the main predictions and effects. The paper would be much stronger if the authors could integrate their main predictions into a theory (lines 146, 150). The theory of ‘fitness interdependence’ might offer such background.
Fitness interactions can be very diverse. Perhaps the most well-known is the kin selection, where the interaction is due to genetic relatedness. Famously, Hamilton's rule (Hamilton, 1963, 1964) describes the effect of such relatedness on the evolution of cooperation. This idea, however, can be generalized beyond kin selection. There are many implementations and terms used for this effect: ‘pseudo-reciprocity’ (Connor, 1986), ‘group augmentation’ (Kokko et al., 2001). Long term cooperation can lead to this effect as well (Tooby & Cosmides, 1996). Roberts (2005) termed these pay-off interactions between unrelated individuals as ‘stake’. Some called it 'vested interests' see (Barclay & Van Vugt, 2015). A more widespread term is ‘fitness interdependence’ (Aktipis et al., 2018; Cronk et al., 2018). There is a steadily growing number of papers explaining the evolution of cooperation or signalling using the framework of fitness interdependence (Barclay, 2020; Barclay et al., 2021). The RQ factor termed ‘Relationship Value’ denotes a positive fitness interaction. Relationship Value is clearly influenced by kinship (as expected based on the above models) and higher Relationship Value predicts higher DCT - again, as expected- because higher Relationship value implies higher ‘stake’ or higher ‘vested’ interest in the survival of the other individual thus it should promote stronger cooperation between the partners. Thus one of the main predictions of the paper can be derived from this theory.
Comments:
"Feeding tolerance is therefore considered an important driver in the evolution of human food sharing1" line 48
Is not feeding tolerance is the result of increased food sharing? How do we know?
"Incompatibility, is a measure of the general nature of the interactions between the two partners." Line 87
What does it mean? It would be nice to have an example.
"Security, describes the predictability or consistency of interactions between partners over time." Line 88
Why does it called security if it means consistency? Also, does consistency imply that the frequency of interactions over time is the same, or does it imply the same type of interactions (i.e. antagonistic vs. cooperative)?
I think that details of the RQ system would better fit into a method section, while the intro should focus on the main questions (see below).
"We expect that bonobos do not systematically differ from chimpanzees" line 146
Why? Is there any theoretical justification for this prediction? Are they using the same type of resource, are they showing the same type of feeding patterns in nature?
"We expect to find an association between measures of Dyadic Cofeeding Tolerance and RQ similar to previous studies8,25,26,36,54, with higher cofeeding tolerance present in dyads with higher Relationship Values and lower Incompatibilities. 153" line 150.
Why? Is there a theoretical justification for this? I think there is (see fitness interdependence).
The above two questions are the main questions of the study, they would deserve a more detailed explanation in the intro (instead of the details of the RQ system).
"The universal effect of social bond strength on feeding tolerance across the two Pan species indicates that this association is evolutionarily conserved and most likely present in the last common ancestor shared between humans and Pan." Line 652.
I think that this conclusion would be valid only if the same results (same effects as in this study) would have been derived for humans as well.
References
Aktipis, A., Cronk, L., Alcock, J., Ayers, J. D., Baciu, C., Balliet, D., Boddy, A. M., Curry, O. S., Krems, J.A., Muñoz, A., Sullivan, D., Sznycer, D., Wilkinson, G. S., & Winfrey, P. (2018). Understanding cooperation through fitness interdependence. Nature Human Behaviour, 2(7), 429–431. https://doi.org/10.1038/s41562-018-0378-4
Barclay, P. (2020). Reciprocity creates a stake in one’s partner, or why you should cooperate even when anonymous. Proceedings of the Royal Society of London B, 287, 20200819. https://doi.org/10.1098/rspb.2020.0819
Barclay, P., & Van Vugt, M. (2015). The evolutionary psychology of human prosociality: adaptations, mistakes, and byproducts. In D. Schroeder & W. Graziano (Eds.) Oxford Handbook of Prosocial Behavior, pp. 37-60. Oxford, UK: Oxford University Press.
Barclay, P., Bliege Bird, R., Roberts, G., & Számadó, S. (2021). Cooperating to show that you care: costly helping as an honest signal of fitness interdependence. Philosophical Transactions of the Royal Society B, 376(1838), 20200292.
Connor RC. 1986 Pseudo-reciprocity: investing in mutualisms. Anim. Behav. 34, 1562–1584. (doi:10. 1016/S0003-3472(86)80225-1)
Cronk L, Steklis D, Steklis N, Akker OR Van Den, Aktipis A. 2018 Kin terms and fitness interdependence. Evol. Hum. Behav. 40, 281–291. (doi:10.1016/j.evolhumbehav.2018.12.004)
Hamilton, W. D. (1963). "The evolution of altruistic behavior". American Naturalist. 97 (896): 354–356.
Hamilton, W.D. (1964). "The Genetical Evolution of Social Behaviour. II". Journal of Theoretical Biology. 7 (1): 17–52.
Kokko H, Johnstone RA, Clutton-Brock TH. 2001 The evolution of cooperative breeding through group augmentation. Proc. R. Soc. Lond. B 268, 187–196. (doi:10.1098/rspb.2000.1349)
Roberts, G. (2005). Cooperation through interdependence. Animal Behaviour, 70, 901-908.
Tooby J, Cosmides L. 1996 Friendship and the Banker’s paradox: other pathways to the evolution of adaptations for altruism. Proc. Br. Acad. 88, 119–143.
Author Response
Response to Reviewer 2 Comments
Point 1: The authors investigate Dyadic Cofeeding Tolerance in bonobos and chimpanzees using the “resource plot” experimental paradigm. The authors present a methodological innovation by constructing a novel Relationship Quality index. The study shows that DCT affected in the same way in both species. Finally, the authors conclude that this effect seems to be evolutionarily conserved and likely to be present in the common ancestor of humans and chimpanzees. My main problem is the lack of theoretical background when deriving the main predictions and effects. The paper would be much stronger if the authors could integrate their main predictions into a theory (lines 146, 150). The theory of ‘fitness interdependence’ might offer such background. Fitness interactions can be very diverse. Perhaps the most well-known is the kin selection, where the interaction is due to genetic relatedness. Famously, Hamilton's rule (Hamilton, 1963, 1964) describes the effect of such relatedness on the evolution of cooperation. This idea, however, can be generalized beyond kin selection. There are many implementations and terms used for this effect: ‘pseudo-reciprocity’ (Connor, 1986), ‘group augmentation’ (Kokko et al., 2001). Long term cooperation can lead to this effect as well (Tooby & Cosmides, 1996). Roberts (2005) termed these pay-off interactions between unrelated individuals as ‘stake’. Some called it 'vested interests' see (Barclay & Van Vugt, 2015). A more widespread term is ‘fitness interdependence’ (Aktipis et al., 2018; Cronk et al., 2018). There is a steadily growing number of papers explaining the evolution of cooperation or signalling using the framework of fitness interdependence (Barclay, 2020; Barclay et al., 2021). The RQ factor termed ‘Relationship Value’ denotes a positive fitness interaction. Relationship Value is clearly influenced by kinship (as expected based on the above models) and higher Relationship Value predicts higher DCT - again, as expected- because higher Relationship value implies higher ‘stake’ or higher ‘vested’ interest in the survival of the other individual thus it should promote stronger cooperation between the partners. Thus one of the main predictions of the paper can be derived from this theory.
Response 1: We sincerely thank the reviewer for their helpful suggestions on how to improve the theoretical framework of the manuscript. We have now incorporated the terminology regarding fitness interdependence into the introduction and discuss its relevance in more detail in the discussion. Especially given that we are working with captive populations where environmental conditions are almost identical, the interdependence framework might explain why we do not find differences between the two species as individuals do not rely on each other as heavily for survival and protection of valuable resources from neighbouring communities. We also discuss in length in the discussion what our expectations therefore would be in wild populations, given that they show considerable variation in socio-ecological conditions that have an impact on fitness interdependence of individuals and groups.
Point 2: "Feeding tolerance is therefore considered an important driver in the evolution of human food sharing1" line 48. Is not feeding tolerance is the result of increased food sharing? How do we know?
Response 2: We agree with the reviewer that this can not be concluded with certainty and have deleted this statement from the introduction.
Point 3: "Incompatibility, is a measure of the general nature of the interactions between the two partners." Line 87. What does it mean? It would be nice to have an example.
Response 3: We have added more information regarding this component at the end of the sentence:
Line 102: The second component, Incompatibility, is a measure of the general nature of the interactions between the two partners and typically reflects levels of aggression and counter-intervention during agonistic interactions.
Point 4: "Security, describes the predictability or consistency of interactions between partners over time." Line 88. Why does it called security if it means consistency? Also, does consistency imply that the frequency of interactions over time is the same, or does it imply the same type of interactions (i.e. antagonistic vs. cooperative)?”
Response 4: Given that this information was not crucial for the purpose of this paper and your recommendation in point 5, we have shortened the section on relationship quality methodology and only discuss the two components that are consistently found across the bonobo and chimpanzee studies (Value and Incompatibility).
Point 5: I think that details of the RQ system would better fit into a method section, while the intro should focus on the main questions (see below).
Response 5: Based on this suggestion and one from reviewer three, we have made the section on relationship quality in bonobos and chimpanzees in the introduction more concise and added a table that provides a better overview of the effects of sex, kinship and age on relationship quality components based on previous literature, see table 1.
Point 6: "We expect that bonobos do not systematically differ from chimpanzees" line 146
Why? Is there any theoretical justification for this prediction? Are they using the same type of resource, are they showing the same type of feeding patterns in nature?
"We expect to find an association between measures of Dyadic Cofeeding Tolerance and RQ similar to previous studies8,25,26,36,54, with higher cofeeding tolerance present in dyads with higher Relationship Values and lower Incompatibilities. 153" line 150.
Why? Is there a theoretical justification for this? I think there is (see fitness interdependence).
The above two questions are the main questions of the study, they would deserve a more detailed explanation in the intro (instead of the details of the RQ system).
Response 6: As mentioned above, we are grateful for this helpful suggestion on how to improve our framework and we have reduced the detailed discussion of the RQ system in the introduction and added the fitness interdependence framework instead. Given that we did not design the study specifically to test this theory we have mostly incorporated its relevance for our results to the discussion part of the manuscript. The predictions are now based on previous findings for the two species with short examples of expectations based on fitness interdependence theory.
Point 7: "The universal effect of social bond strength on feeding tolerance across the two Pan species indicates that this association is evolutionarily conserved and most likely present in the last common ancestor shared between humans and Pan." Line 652.
I think that this conclusion would be valid only if the same results (same effects as in this study) would have been derived for humans as well.
Response 7: We have changed the main conclusion of the manuscript to better fit our discussion of the relevance of the fitness interdependence theory and the flexibility and adaptability of Pan.
Line 617: The quality of social bonds also predicted Dyadic Cofeeding Tolerance in an identical manner in both Pan species and provides a mechanism that could explain previously reported between-group variation in feeding tolerance in Pan46,53, as the strength of social relationships can vary between populations depending on demographics54 and ecological conditions 85. Our results indicate that cofeeding tolerance in Pan is flexible and likely regulated through changing levels of mutual interdependence relationships, which are in turn reliant on environmental conditions. This flexibility in tolerance appears to outweigh species-specific physiological restraints on tolerance and highlights the adaptiveness of the two species to changing environments. Our study on zoo-housed chimpanzees and bonobos offers an operational measure to quantify cofeeding tolerance that can be applied to a wider range of populations of chimpanzee and bonobo populations, but also other primate or mammal species, both in the wild and in zoos or sanctuaries to see how variation in local socio-ecological circumstances influences fitness interdependence and cofeeding tolerance on a dyadic and group level. This may ultimately lead to a better understanding of how local environments have shaped the evolution of tolerance in humans and other species.
Reviewer 3 Report
My overall comment is that this is a decent paper, the study is well conducted, the analysis is appropriate, but it is also a very uninteresting paper if you are not already heavily invested in this topic. I would urge you to make the paper broader - I am a primatologist and even I found it way too focussed and too narrow in its scope. If I had to review this for say Folia Primatologica or the International Journal of Primatology I would still have urged you to make it more appealing to a wider audience. You decided to submit it to MDPI Biology, meaning you are aiming for a very wide audience. Take a step back and think what this all means in a broad sense. You start by focussing on humans -- that could be one angle-- but then you do not carry that theme through. But there are other ways to broaden it as well.
Below are some minor comments:
The title is quite long – I wonder if a shorter more focussed title is possible.
Abstract is good, but please include where and when the study was conducted, and thus make it clear this is a zoo-based study.
Line 67. Relationship quality is not too long – I would use this rather than RQ (spell this out throughout the paper)
Line 71. Second, in both species, food sharing is not limited to kin or mates, but also takes place between unrelated group members. How is this different from other primates (e.g., macaques, baboons), or even other vertebrates? Please explain.
Line 76-102– I think this is fairly dense in terms of information and I was wondering if this is possible to summarize in a table where you can directly compare the two species
Line 162 How does the presence of infants and juveniles influence cofeeding tolerance and dyadic relationship quality? Perhaps add a few sentences how this can have an influence (or not) to justify your decision.
Line 168 I would include Table S1 here and not as a supplement. And then include the data from Line 179 (hours of observation) in this Table.
Line 205 – Factors affecting relationship quality needs a 2.2.2.
Line 311 – add that these same variables were used for both species.
Line 382-400 – this is very densely written and difficult to follow (that is not to say it is incorrect, but is there a way to present this in a more descriptive way and then perhaps add some of the statistical details in a table).
Line 497 and further. This is the weakest part of the paper.
In the Introduction you start with “Tolerant food sharing among non-kin is considered a hallmark of human cooperative nature”, and “However, humans belong to one of the few species that also regularly share food outside of the kin or mating context. This raises an evolutionary question, since this type of food sharing does not provide obvious fitness benefits10, but rather results in the loss of a resource or increased competition in the food patch” and “Humans are considered a highly tolerant species, since food is regularly shared without aggression. Feeding tolerance is therefore considered an important driver in the evolution of human food sharing.”
This suggests that one of the main drivers behind this study was to understand more about humans, and that justifies your selection of study species (bonobo and chimpanzee as our closest relatives). But nothing of this comes back in the Discussion of the paper. In fact, the entire Discussion is focussed on your model, how well it worked in your setting, with your study population. I think you have good data and you really should expand on this, and explain, speculate, discuss, contemplate, etc. what your findings mean for a broader set of species including humans.
In addition, I think you need to discuss more the constraints / limitations of your study in terms of captivity, and what we can expect in wild chimpanzees and bonobos. For instance, in chimpanzees most males are closely related but females are not, and sharing among non-kin is therefore more relevant / common perhaps, for females. In zoos I assume the males are not all closely related and this may have an effect on the outcome.
The references are very detailed and correct.
Given that this is an online journal and you do not have a restriction on page length, I would suggest including some of the data that is now in the Supplementary Materials in the main body of the paper.
Author Response
Response to Reviewer 3 Comments
Point 1: My overall comment is that this is a decent paper, the study is well conducted, the analysis is appropriate, but it is also a very uninteresting paper if you are not already heavily invested in this topic. I would urge you to make the paper broader - I am a primatologist and even I found it way too focussed and too narrow in its scope. If I had to review this for say Folia Primatologica or the International Journal of Primatology I would still have urged you to make it more appealing to a wider audience. You decided to submit it to MDPI Biology, meaning you are aiming for a very wide audience. Take a step back and think what this all means in a broad sense. You start by focussing on humans -- that could be one angle-- but then you do not carry that theme through. But there are other ways to broaden it as well.
Response 1: We thank the reviewer for his time and efforts towards giving us suggestions for improvement of our manuscript. We have taken this commentary into account and combined with the suggestions of reviewer two, we have now added the theoretical framework of fitness interdependence to the introduction and discussion of the manuscript to appeal to a wider audience. Using this theoretical framework we now discuss how the lack of a difference in tolerance between the two species might be due to their captive circumstances and what our expectations would be in wild populations. We discuss these things in more detail as an answer to your points below.
Point 2: The title is quite long – I wonder if a shorter more focussed title is possible.
Response 2: We have shortened the title to “Drivers of Dyadic Cofeeding Tolerance in Pan: A composite measure approach”
Point 3: Abstract is good, but please include where and when the study was conducted, and thus make it clear this is a zoo-based study.
Response 3: We have added this to the abstract
Point 4: Line 67. Relationship quality is not too long – I would use this rather than RQ (spell this out throughout the paper)
Response 4: We have changed this throughout the manuscript
Point 5: Line 71. Second, in both species, food sharing is not limited to kin or mates, but also takes place between unrelated group members. How is this different from other primates (e.g., macaques, baboons), or even other vertebrates? Please explain.
Response 5: We have added a statement on the rare nature of this behavior in other primate species to this section along with a review by Jaeggi and Gurven on tolerance in different primate species.
Line 87: Similar to humans, food sharing is not limited to kin or mates, but also takes place between unrelated group members6,10,36–38, which is rare among primates and makes bonobos and chimpanzees good models to investigate the impact of both kin and non-kin fitness interdependency interactions on tolerance39.
Point 6: Line 76-102– I think this is fairly dense in terms of information and I was wondering if this is possible to summarize in a table where you can directly compare the two species.
Response 6: We have shortened this section and added a table (table 1) that sums up the details of the previous studies. We now focus solely on the two components that are consistently found across studies for both species (Value and Incompatibility) that are most relevant for our study.
Point 7: Line 162 How does the presence of infants and juveniles influence cofeeding tolerance and dyadic relationship quality? Perhaps add a few sentences how this can have an influence (or not) to justify your decision.
Response 7: While the presence of infants and juveniles could impact both cofeeding tolerance and dyadic relationship quality, the nature of their social relationships is markedly different from that between mature individuals. Following previous relationship quality studies, we therefore decided to exclude them as especially for RQ analysis the inclusion of their immature social relationships is expected to make a difference to the final factor solution. Considering their levels of grooming given and involvement in aggressive interactions are rather low compared to those of mature individuals this would bias the results. We have added this to the methodology.
Line 166: All groups contained infants and/or juveniles (< 7 years) but following the methodology of previous studies on relationship quality these were excluded from analysis given that the nature of their social relationships substantially differs from those of mature individuals, especially when considering levels of grooming and agonistic behaviour 29–31.
Point 8: Line 168 I would include Table S1 here and not as a supplement. And then include the data from Line 179 (hours of observation) in this Table.
Response 8: We have moved the table from the supplement to the main body of the manuscript and added the information of observational data to the table, see Table 2.
Point 9: Line 205 – Factors affecting relationship quality needs a 2.2.2.
Response 9: We have adjusted the numbering throughout the manuscript
Point 10: Line 311 – add that these same variables were used for both species.
Response 10: We have added in the methods section that all variables were scored identically in both species
Point 11: Line 382-400 – this is very densely written and difficult to follow (that is not to say it is incorrect, but is there a way to present this in a more descriptive way and then perhaps add some of the statistical details in a table).
Response 11: We have put the statistical information in a table (table 5) leaving only p-values for signififcant effects throughout the text and changed the wording to make this less dense. We did the same for the paragraph on factors affecting DCT, see point 3.4.2.
Line 354: No significant species by sex combination interaction effect was found on either component and component scores also did not differ between species or with age difference of the dyad (Table 5, Figure S1). Instead, Relationship Value and Incompatibility were significantly affected by kinship and sex combination of the dyad in a similar fashion in both species (Figure 1). Dyads that were maternal kin scored higher on Relationship Value (p = 0.014) and lower on Relationship Incompatibility (p < 0.001) compared to other dyads (Figure 1). Female-female dyads scored higher on Relationship Value (p = 0.015) and lower on Relationship Incompatibility (p < 0.001) than female-male dyads (Figure 1). Compared to male-male dyads, female-female dyads also scored lower on Relationship Incompatibility (p < 0.001) but not on Relationship Value (p = 0.636). Finally, female-male dyads did not differ from male-male dyads in either dimension (Table 5).
Point 12: Line 497 and further. This is the weakest part of the paper.In the Introduction you start with “Tolerant food sharing among non-kin is considered a hallmark of human cooperative nature”, and “However, humans belong to one of the few species that also regularly share food outside of the kin or mating context. This raises an evolutionary question, since this type of food sharing does not provide obvious fitness benefits10, but rather results in the loss of a resource or increased competition in the food patch” and “Humans are considered a highly tolerant species, since food is regularly shared without aggression. Feeding tolerance is therefore considered an important driver in the evolution of human food sharing.” This suggests that one of the main drivers behind this study was to understand more about humans, and that justifies your selection of study species (bonobo and chimpanzee as our closest relatives). But nothing of this comes back in the Discussion of the paper. In fact, the entire Discussion is focussed on your model, how well it worked in your setting, with your study population. I think you have good data and you really should expand on this, and explain, speculate, discuss, contemplate, etc. what your findings mean for a broader set of species including humans.
Response 12: Using the suggestions from reviewer two and three we have now made adjustments to the final paragraphs of the discussion. The main purpose of this paper was to provide a novel analytical approach that can be used in both wild and captive settings to measure cofeeding tolerance in a variety of species. We have put this more clearly in the discussion along with its implications for use in other species and relevance for studying the evolution of (human) tolerance.
Line 617: The quality of social bonds also predicted Dyadic Cofeeding Tolerance in an identical manner in both Pan species and provides a mechanism that could explain previously reported between-group variation in feeding tolerance in Pan46,53, as the strength of social relationships can vary between populations depending on demographics54 and ecological conditions 85. Our results indicate that cofeeding tolerance in Pan is flexible and likely regulated through changing levels of mutual interdependence relationships, which are in turn reliant on environmental conditions. This flexibility in tolerance appears to outweigh species-specific physiological restraints on tolerance and highlights the adaptiveness of the two species to changing environments. Our study on zoo-housed chimpanzees and bonobos offers an operational measure to quantify cofeeding tolerance that can be applied to a wider range of populations of chimpanzee and bonobo populations, but also other primate or mammal species, both in the wild and in zoos or sanctuaries to see how variation in local socio-ecological circumstances influences fitness interdependence and cofeeding tolerance on a dyadic and group level. This may ultimately lead to a better understanding of how local environments have shaped the evolution of tolerance in humans and other species.
Point 13: In addition, I think you need to discuss more the constraints / limitations of your study in terms of captivity, and what we can expect in wild chimpanzees and bonobos. For instance, in chimpanzees most males are closely related but females are not, and sharing among non-kin is therefore more relevant / common perhaps, for females. In zoos I assume the males are not all closely related and this may have an effect on the outcome.
Response 13: We thank the reviewer for this suggestion and have incorporated this into the discussion. Using the fitness interdependence framework we now create clear expectations based on socio-ecological differences in the wild that are likely absent in our captive populations.
Line 582: In conclusion, our results show that bonobos and chimpanzees do not differ consistently in Dyadic Cofeeding Tolerance, which supports earlier findings of within-species variation in cofeeding tolerance being larger than between-species variation in bonobos and chimpanzees46. Important to note, is that the lack of clear species differences in Dyadic Cofeeding Tolerance might be attributable to the reduced importance of fitness interdependence in captive Pan populations due to more similar socio-ecological conditions versus wild populations. In the wild, bonobos and chimpanzees are both male philopatric species, and experience similar predation risks84,85. However resource distribution and food availability are believed to be responsible for differences in between- and within-group competition and levels of kin-driven fitness interdependence 84,85. Generally speaking, in chimpanzees, seasonality is higher and resources are more clumped, which results in higher competition between females, who each occupy and compete for high quality core areas, and show lower tolerance towards immigrating non-kin females, and commit female infanticide86,87. Chimpanzee males are related because of male philopatry and form largely kin-driven alliances and strong social bonds to defend a communal territory with its resident females against males from neighbouring communities. This results in strong between-group competition, including border patrols, intercommunity killing, raids and infanticide88–90. In bonobos, seasonality in food availability is generally considered to be lower and resources are less clumped, resulting in lower between-and within-group competition42,91. Relationships between bonobo communities are more relaxed, no intercommunity killing or infanticide has been described, and resident bonobo females are less hostile towards immigrating females92. Bonobo females form alliances with non-related females, and form coalitions that allow for female co-dominance42,93. While male bonobos are related to each other because of male philopatry, similar to chimpanzee males, bonds between male bonobos are weak and they do not form coalitions or border patrols71,72. This shows that compared to chimpanzees, fitness interdependence in wild bonobos is less kin-driven. At a group level, this is expected to result in higher fitness interdependence in wild bonobo populations, and therefore higher levels of overall group cofeeding tolerance compared to wild chimpanzee populations. Despite this reported species duality, bonobos and chimpanzees seem to show behavioural plasticity. Western chimpanzee populations seem to experience less food seasonality, resulting in more relaxed relationships between females compared to eastern populations94, and also within wild bonobo communities between-group competition increases when food abundance is lower95, and some populations seem to experience more seasonality in food availability but how this relates to tolerance within and between groups is unknown96. This flexibility within Pan species likely explains why our study, as well as other captive work, do not find the predicted species differences in cofeeding tolerance, as resource distribution in captive populations is similar and intergroup competition is lacking which reduces the need for male alliance formation in zoo-housed chimpanzees.
Point 14: The references are very detailed and correct. Given that this is an online journal and you do not have a restriction on page length, I would suggest including some of the data that is now in the Supplementary Materials in the main body of the paper.
Response 14: We thank the reviewer for his detailed review. We have moved some of the information to the main body of the text and keep only the less relevant tables and results in the supplement as to keep focus on the important results in the main body of the text.
Reviewer 4 Report
The authors present an interesting study on social bonds and dyadic cofeeding tolerance on captive bonobos and chimps. The paper is generally well written and the data analysis is sound, but there are important issues to be fixed before this paper can be considered for publication.
1. Missing the simple summary as per Biology guidelines.
2. At line 39. Which question? Not clear. This evolutionary importance of your study is present in several places but not well expanded so it is not clear to the reader how your findings can be of any use for human evolution. This is a major issue that needs to be fixed with additions to simple summary, abstract, introduction, discussion, and conclusion.
3. You should report the version of R used not R Studio as the latter is just an interface to R software.
4. I do not like that data collection and analysis is in the same place, it is a bit confusing. You can also avoid repetition of some sentences if you have data analysis all in one place (now methods are a bit long and sometimes repetitive).
5. Ethical statement should go in the dedicated section "Institutional Review Board Statement" after the conclusion
Author Response
Response to Reviewer 4 Comments
Point 1: The authors present an interesting study on social bonds and dyadic cofeeding tolerance on captive bonobos and chimps. The paper is generally well written and the data analysis is sound, but there are important issues to be fixed before this paper can be considered for publication. 1. Missing the simple summary as per Biology guidelines.
Response 1: We thank the reviewer for this positive feedback on our submission and have added the simple summary to the new version.
Point 2: At line 39. Which question? Not clear. This evolutionary importance of your study is present in several places but not well expanded so it is not clear to the reader how your findings can be of any use for human evolution. This is a major issue that needs to be fixed with additions to simple summary, abstract, introduction, discussion, and conclusion.
Response 2: We have added this using suggestions from reviewers 2 and 3. We have now added the theoretical framework of fitness interdependence to the introduction and discussion and have expanded more on the usage of our proposed analytical approach to measure tolerance in a variety of species and settings to further investigate how tolerance fluctuates with changing socio-ecological circumstances. This can be seen in the simple summary, abstract and final part of the discussion
Point 3: You should report the version of R used not R Studio as the latter is just an interface to R software.
Response 3: We have adjusted this in the methods.
Point 4: I do not like that data collection and analysis is in the same place, it is a bit confusing. You can also avoid repetition of some sentences if you have data analysis all in one place (now methods are a bit long and sometimes repetitive).
Response 4: We have separated these sections to avoid repetition of analyses.
Point 5: Ethical statement should go in the dedicated section "Institutional Review Board Statement" after the conclusion
Response 5: We have moved this to the correct section.
Round 2
Reviewer 3 Report
You have done a good job in addressing my comments, which were fairly general, and that of the other reviewer. I have no more comments or suggestions
Author Response
We thank the reviewer for his positive recommendation of our resubmission
Reviewer 4 Report
The authors did a good job addressing my comments and the comments of the other reviewers. I have a few more suggestions:
- Line 23-25. better to remove the sentence from the simple summary as it is repeated in the abstract
- Line 45. I am not sure that should is the right word here
- I might be wrong, please check, but the style of numbered references is not correct. Instead of the superscript you should use square brackets, e.g. [1,2]
- Table 1. You should also explain what the symbols > and < means (they are obvious but still you should explain them, also the symbols for positive and negative effects can be + and -)
- When you talk about Pan I would consider adding spp., or maybe change the wording in some points. For example, in line 127 it is a bit weird to refer to resulting Pan relationship quality model, what is a Pan model/ Pan relationship quality model? A reader can understand what you mean, but maybe you can edit a bit the sentences to make it clearer. Maybe explain the first time and there refer to it as Pan model or something like that, e.g. using hereafter Pan model
- Line 195. Now it is data collection only
- Line 235. superscript for m2
- Line 353. R version instead of R studio, and can even repeat the software and package once as you use the same for LMM and GLMM
- Line 480. Missing content in brackets (I guess p-value).
- Line 643. It is better to have a section "conclusion" as per MDPI guidelines (when the discussion is long), so I would just have a section "Conclusion" and remove "In conclusion,"
Author Response
Response to Reviewer 4 Comments
Point 1: The authors did a good job addressing my comments and the comments of the other reviewers. I have a few more suggestions: Line 23-25. better to remove the sentence from the simple summary as it is repeated in the abstract
Response 1: We are happy to have received a positive recommendation for our resubmission and thank the reviewer for his efforts towards improving the quality of the manuscript. We have removed this sentence from the simple summary
Point 2: Line 45. I am not sure that should is the right word here.
Response 2: We have changed this to “can”.
Point 3: I might be wrong, please check, but the style of numbered references is not correct. Instead of the superscript you should use square brackets, e.g. [1,2]
Response 3: Thank you for pointing this out, we have changes this throughout the manuscript
Point 4: Table 1. You should also explain what the symbols > and < means (they are obvious but still you should explain them, also the symbols for positive and negative effects can be + and -)
Response 4: We have added this to the table
Point 5: When you talk about Pan I would consider adding spp., or maybe change the wording in some points. For example, in line 127 it is a bit weird to refer to resulting Pan relationship quality model, what is a Pan model/ Pan relationship quality model? A reader can understand what you mean, but maybe you can edit a bit the sentences to make it clearer. Maybe explain the first time and there refer to it as Pan model or something like that, e.g. using hereafter Pan model
Response 5: We have added this to line 123
Point 6: Line 195. Now it is data collection only, Line 235. superscript for m2, Line 353. R version instead of R studio, and can even repeat the software and package once as you use the same for LMM and GLMM, Line 480. Missing content in brackets (I guess p-value), Line 643. It is better to have a section "conclusion" as per MDPI guidelines (when the discussion is long), so I would just have a section "Conclusion" and remove "In conclusion,"
Response 6: We have made these minor edits to the manuscript.